

# Improving maize carbon and nitrogen metabolic pathways and yield with nitrogen application rate and nitrogen forms

Fanhao Meng[1,2], Ruifu Zhang[1,2], Yuqin Zhang[1,2], Weimin Li[1,2], Yushan Zhang[1,2], Mingwei Zhang[3], Xuezhen Yang[1,2] and Hengshan Yang[1,2]

[1] College of Agronomy, Inner Mongolia Minzu University, Tongliao, Inner Mongolia, China
[2] Inner Mongolia Autonomous Region Feed Crop Engineering Technology Research Center, Inner Mongolia Minzu University, Tongliao, Inner Mongolia, China
[3] Xingan League Institute of Agriculture and Animal Husbandry Science, Wulanhaote, Inner Mongolia, China

Corresponding author
Hengshan Yang,
yanghengshan2003@aliyun.com

## ABSTRACT

Reduced fertilizer efficiency caused by excessive use of nitrogen (N) fertilizer is a major problem in agriculture and a hot topic of research. Most studies have focused on the effect of N application rate on N efficiency, whereas there are limited studies on changing the N form to improve N yield and efficiency. Here, the effects of different N application rates and nitrate-to-ammonium N ratios on post-anthesis carbon (C) and N metabolism and maize yield under shallow-buried drip irrigation were investigated. Two rates of N application (210 kg·ha$^{-1}$ (NA1) and 300 kg·ha$^{-1}$ (NA2)) and three nitrate-to-ammonium N ratios (2:1 (NF1), 3:1 (NF2), and 4:1 (NF3)) were utilized. Post-anthesis photosynthetic characteristics, activities of key enzymes in photosynthetic C and N metabolism, nonstructural carbohydrate content, post-anthesis N accumulation and transportation, yield, and N-use efficiency were determined. At both N application rates, NF2 treatment enhanced photosynthetic activity in the ear-leaf at silking stage and promoted key enzyme activities of C and N metabolic pathways, compared with NF1 and NF3. Furthermore, NF2 significantly increased nonstructural carbohydrate accumulation (4.00–64.71%), post-anthesis N accumulation and transportation (11.00–38.00%), and grain yield (2.60–13.08%). No significant differences between NA1 and NA2 were observed under NF2 in most of the measured variables; however, NA1 had higher N-use efficiency. Thus, the optimal treatment under shallow-buried drip irrigation was a N application rate of 210 kg ha$^{-1}$ and a nitrate-to-ammonium N ratio of 3:1. These findings provide theoretical guidance on appropriate N applications for high-yield maize production.

## INTRODUCTION

As the primary factor promoting plant growth (*Dobermann & Cassman, 2002*), N fertilizers are often applied to crops to pursue high yield and ensure global food security (*Tilman et al., 2001*). However, excessive application of N fertilizer results in a decline in N efficiency year by year (*Guo et al., 2010*; *Liang et al., 2011*; *Liu et al., 2013*). The phenomenon that increasing N fertilizer does not increase yield has recently become prominent in the corn production of China's West Liaohe Plain. This area forms part of the golden corn belt—one of the regions with the highest potential for spring corn production in China (*Zhang et al., 2011*). Consequently, synergistic improvement of yield and N efficiency in spring corn production in this area is an important issue.

Studies on improving N efficiency have predominantly focused on N application rate (*Chong et al., 2022*; *Gu et al., 2017*). However, it has been reported that different N forms can promote photosynthesis by changing the two basic physiological metabolisms of plants —carbon (C) and N metabolism—and ultimately improve yield and N efficiency (*Duan et al., 2007*; *Wang et al., 2019a*, *2021b*). The photosynthetic performance of wheat (*Triticum aestivum*) (*Fu et al., 2022*), rice (*Oryza sativa*) (*Fu, Cui & Shen, 2021*), and flue-cured tobacco (*Nicotiana tabacum*) (*Huang et al., 2021*) increased when fertilized with a combination of nitrate and ammonium N compared with a single form of N. Traditional topdressing N fertilizers are mainly amide N (urea) (*Gheysari et al., 2015*), whereas nitrate and ammonium are inorganic forms of N that can be directly absorbed and utilized by plants (*Dong et al., 2023*; *Pan, 1979*; *Clawson et al., 2008*; *Qiang et al., 2019*; *Zhang et al., 2023*). Plants mainly complete the metabolism of C and N through photosynthesis. After nitrogen is absorbed by plant roots, different amino acids are formed into protein synthesis through the transfer of glutamic acid or glutamine. Nitrogen supplementation with nitrogen forms of nitrate and ammonium nitrogen can improve the nitrogen absorption rate, thus promoting the production of protein in plants and enhancing the activities of enzymes related to carbon and nitrogen metabolism in plants. Thus, the net photosynthetic rate of plants can be increased, the conversion and utilization of non-structural carbohydrates in plants can be accelerated, and the generation of carbon and nitrogen metabolites can be promoted (*Deng et al., 2023*; *Wang et al., 2022*). There are studies showing that degradation and synthesis activities of sucrose synthase were positively associated with nitrate and ammonium N application rates, respectively (*Li et al., 2003*). In addition, *Huang, Cao & Wang (2019)* showed that combined application of nitrate and ammonium N significantly increased carbohydrate accumulation and 3:1 was the optimal nitrate-to-ammonium N ratio.

The appropriate nitrate-to-ammonium N ratio for a specific crop depends on the growth habit of the crop (*Yang et al., 2022b*; *Wang et al., 2007*). Maize is a photophilous $C_4$ plant (*Mueller & Vyn, 2016*), and the leaf C and N metabolism processes differ from those of most $C_3$ plants (*Li & Yan, 2013*; *Yang et al., 2022a*; *Wang et al., 2021a*). Therefore, there is value in examining the changes of C and N metabolism, photosynthesis and photosynthate production, and transformation of maize after application of different proportions of nitrate and ammonium N. This study aimed to explore such changes in

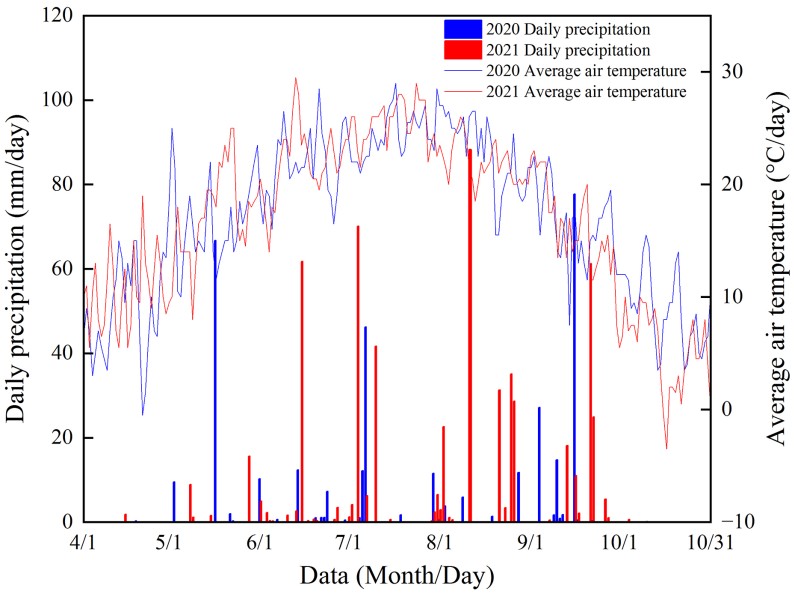

**Figure 1 Precipitation in the maize growing season at the study site during the experimental period.**

maize in China's West Liahoe Plain. The hypothesis of the study was that application of a suitable combination of nitrate and ammonium N would change C and N metabolism in maize, thereby improving maize photosynthesis and ultimately achieving synergistic improvement of maize N content, yield, and N efficiency. Determining the suitable amount of N fertilizer in China's West Liaohe Plain corn belt would provide a reference for corn production in similar areas.

# MATERIALS AND METHODS

## Study site

Field trials were conducted at the Tongliao City New Maize Technology Integration and Promotion Demonstration Base, Horqin District, Tongliao City, China (43°44′N, 122°12′ E) in 2020 and 2021. The study site has an annual average air temperature of 6.6 °C, an active accumulated temperature (≥10 °C) of 3,200 °C, and an average annual frost-free period of 154 d. Figure 1 shows the precipitation and air temperature in the maize growing season during the experimental period. The soil at the study site was gray semi-hydromorphic meadow soil. Background soil nutrients in the cultivated layer (0–20 cm depth) before the trials were organic matter 14.71 g·kg$^{-1}$, alkali-hydrolyzable N 53.45 mg·kg$^{-1}$, available phosphorus 9.82 mg·kg$^{-1}$, and readily available potassium 108.51 mg·kg$^{-1}$.

## Experimental design

In the field trials, maize was cultivated using a drip irrigation technique with water–fertilizer integration. A split-spot design was adopted. The main treatment was N application rate at two levels: 210 (NA1) and 300 kg·ha$^{-1}$ (NA2). The auxiliary treatment was nitrate-to-ammonium N ratio at three levels: 2:1 (NF1), 3:1 (NF2), and 4:1 (NF3).

**Table 1 Topdressing fertilizer schemes for maize at different stages of development in different fertilizer treatments.**

| Treatment | 8-leaf age | | 12-leaf age | | Silking stage | |
|---|---|---|---|---|---|---|
| | Calcium ammonium nitrate | Ammonium sulfate | Calcium ammonium nitrate | Ammonium sulfate | Calcium ammonium nitrate | Ammonium sulfate |
| | kg·ha$^{-1}$ | | | | | |
| NA1NF1 | 257.14 | 75.26 | 514.29 | 150.52 | 85.71 | 25.09 |
| NA1NF2 | 289.29 | 51.74 | 578.57 | 103.48 | 96.43 | 17.25 |
| NA1NF3 | 308.57 | 37.63 | 617.14 | 75.26 | 102.85 | 12.54 |
| NA2NF1 | 385.71 | 112.89 | 771.43 | 225.78 | 128.57 | 37.63 |
| NA2NF2 | 433.93 | 77.61 | 867.85 | 155.23 | 144.64 | 25.87 |
| NA2NF3 | 462.85 | 56.45 | 925.71 | 112.89 | 154.29 | 18.82 |

Note:
Treatments: NA1NF1, N application rate of 210 kg·ha$^{-1}$ and nitrate-to-ammonium N ratio of 2:1; NA1NF2, N application rate of 210 kg·ha$^{-1}$ and nitrate-to-ammonium N ratio of 3:1; NA1NF3, N application rate of 210 kg·ha$^{-1}$ and nitrate-to-ammonium N ratio of 4:1; NA2NF1, N application rate of 300 kg·ha$^{-1}$ and nitrate-to-ammonium N ratio of 2:1; NA2NF2, N application rate of 300 kg·ha$^{-1}$ and nitrate-to-ammonium N ratio of 3:1; NA2NF3, N application rate of 300 kg·ha$^{-1}$ and nitrate-to-ammonium N ratio of 4:1.

The control did not receive N treatment. There were three replicate 96 m$^2$ (20 m × 4.8 m) plots for each treatment and the control, for a total of 21 plots. Each treatment was subjected to basal application of diammonium phosphate (N–P–K = 15–46–0) and potassium sulfate (N–P–K = 0–0–50) fertilizers at rates of 210 kg·ha$^{-1}$ and 90 kg·ha$^{-1}$, respectively, and topdressing of calcium ammonium nitrate (NO$_3$N = 14%, NH$_4^+$-N = 1%) and ammonium sulfate (NH$_4^+$-N = 20.5%) as N donors. Topdressing N fertilizers were dissolved in water and applied *via* the drip irrigation system. Each fertilizer was applied in three doses in portions of 3:6:1 at the shooting, big-flare, and silking stages, respectively. The topdressing scheme is summarized in Table 1. The total irrigation amount was 2,400 m$^3$·ha$^{-1}$, which was divided into seven doses that were applied throughout maize growth and development according to the local practice. The maize cultivar used in the trials was 'Jingke 968', which was planted in rows with alternate wide (80 cm) and narrow (40 cm) spacings and at a density of 75,000 plants ha$^{-1}$. Trials were conducted in 2020 and 2021. Maize was seeded at the beginning of May and harvested at the beginning of October.

### Measured variables and methods
#### Photosynthetic characteristics
In the silking stage, ten adjacent plants with growth conditions that were representative of average plot-wide growth conditions were selected from each plot. The photosynthetic physiological indicators of ear leaf in different treatment were determined at time period of 9:00 to 11:00 am by using a LI-6400 XT portable photosynthetic set (LI-COR, Lincoln, NE, USA) at saturated PAR intensity of 1,500 μmol·m$^{-2}$·s$^{-1}$, T leaf 25 °C and carbon dioxide flow rate 500 μmol·m$^{-2}$·s$^{-1}$ on a sunny day at the flowering stage: net photosynthetic rate (P$_n$), intercellular CO$_2$ concentration (C$_i$), transpiration rate (T$_r$), and photosynthetically

active radiation (PAR). Instantaneous light energy utilization efficiency (LEUE), instantaneous water utilization efficiency (WUE), and carboxylation rate were calculated as follows:

Instantaneous LEUE (%) = $P_n$/PAR;

Instantaneous WUE (μmol·mmol$^{-1}$) = $P_n$/$T_r$;

Instantaneous carboxylation rate (mol·m$^{-2}$·s$^{-1}$) = $P_n$/$C_i$.

### Activities of enzymes involved in photosynthetic carbon and nitrogen metabolic pathways

Five adjacent plants with similar growth conditions that were representative of the average plot-wide growth conditions were sampled from each plot every 20 days after anthesis (DAA) to the maturity stage; samples were collected on days 0 (DAA0), 20 (DAA20), 40 (DAA40), and 60 (DAA60). Ear leaves were cut from sample plants, the surfaces were cleaned, veins were removed, and then the leaves were frozen in liquid N and stored at −80 °C. Enzyme-linked immunosorbent assay kits (Shanghai Enzyme Linked Biotechnology Co., Ltd., Shanghai, China) were used to determine activities of ribulose-1,5-bisphosphate carboxylase (RuBisCo), pyruvate phosphate dikinase (PPDK), malic enzyme (ME), malate dehydrogenase (MDH), phosphoenolpyruvate carboxylase (PEPC), nitrate reductase (NR), glutamate synthetase (GOGAT), glutamine synthetase (GS), and glutamate dehydrogenase (GDH). The sample to be tested was diluted by 50 μL for 5 times, reacted with 100 μL enzyme-labeled reagent, warmed and washed, and then added 100 μL color developer to hide from light for 15 min at 37 °C, and added 50 μL termination solution to terminate the reaction. After 15 min, the absorbance (OD) of each sample at 450 nm wavelength was measured by zero adjustment in blank group.

### Content of nonstructural carbohydrates

The sampling scheme and methods were identical to those described in the previous section (*Activities of enzymes involved in photosynthetic carbon and nitrogen metabolic pathways*). Sucrose content was determined using the method of *Qi et al. (2003)*, soluble sugar content was determined using the method of *Li (2000)*, and starch content was determined using the method of *Men & Liu (1995)*.

### Accumulation and remobilization of nitrogen

From 2020 to 2021 at flowering stage and ripening stage, five adjacent maize plants with the same growth potential and representing the average growth potential of the plot were selected for each treatment. The plants were separated into stem sheath, leaf, cob, bract, and grain and were degreened for 30 min at 105 °C in an oven. Dry matter weight was determined after drying at 80 °C to a constant weight, and N content in the different organs was determined after grinding and sifting. Nitrogen accumulation and transportation after silking were calculated according to the formulae:

Nitrogen accumulation (kg·ha$^{-1}$) = Dry matter accumulation at ripening stage × Plant nitrogen concentration at ripening stage (%);

N remobilization (kg·ha$^{-1}$) = N accumulation in the stalk at maturity/N accumulation in the stalk at silking stage;

Partial fertilizer productivity of nitrogen (PFPN, kg·kg$^{-1}$) = Grain yield/Nitrogen application;

Nitrogen fertilizer agronomic use efficiency (NAE, kg·kg$^{-1}$) = (Grain yield in N-application zone – Grain yield in N-free zone)/Nitrogen application amount;

### Maize yield and its components

Maize yield was estimated at the maturity stage. In each plot, yield was estimated in a 24-m$^2$ area. The effective number of ears was surveyed, and yield was estimated based on the actual harvest; this process was repeated three times. Ten fruit ears were randomly selected from each plot, naturally dried, and threshed to measure water content, number of grains per ear, and 1,000-grain weight (1,000 randomly selected grains); this process was repeated three times. The yield at standard water content (14%) was calculated based on the data described above.

## Data treatment and analysis

Data were sorted and tabulated using Excel 2016 (Microsoft, Seattle, WA, USA), and graphs were generated using Origin 2022b (OriginLab, Northampton, MA, USA). All data were analyzed with IBM SPSS Statistics 19.0 (IBM Corp., Armonk, NY, USA). Tukey's honestly significant difference (HSD) test was used to determine the significance of differences between means ($P < 0.05$).

## RESULTS

### Effects of nitrate-to-ammonium nitrogen ratio on maize photosynthetic characteristics

Nitrogen application and nitrate-to-ammonium N ratio had extremely significant effects on maize photosynthetic characteristics, and the interaction between the two factors was significant. In NA1 and NA2, 2-year average silking-stage ear-leaf photosynthetic rate, instantaneous LEUE, instantaneous WUE, and instantaneous carboxylation rate were significantly higher in NF2 than in NF1 by 41.68%, 40.57%, 39.08%, 35.00% and 13.49%, 21.83%, 15.25%, and 12.00% respectively (Fig. 2). For NA1, photosynthetic indicators were significantly higher in NF2 than in NF3 and significantly higher in NF3 than in NF1. For NA2, the instantaneous carboxylation rate was significantly higher in NF2 than in NF3, and the instantaneous LEUE was significantly higher in NF3 than in NF1. For a given nitrate-to-ammonium N ratio, only the photosynthetic rate, instantaneous LEUE, instantaneous WUE, and instantaneous carboxylation rate in NF2 did not differ significantly between NA1 and NA2.

### Effects of nitrate-to-ammonium nitrogen ratio on maize post-anthesis carbon metabolism enzyme activities

Activities of enzymes involved in maize post-anthesis ear-leaf photosynthetic C metabolism decreased during growth and development in all treatments (Fig. 3). For a given N application rate, the post-anthesis C metabolism indicators were significantly higher in NF2 than in NF1 in both years. Average RuBisCo, PPDK, MDH, and PEPC

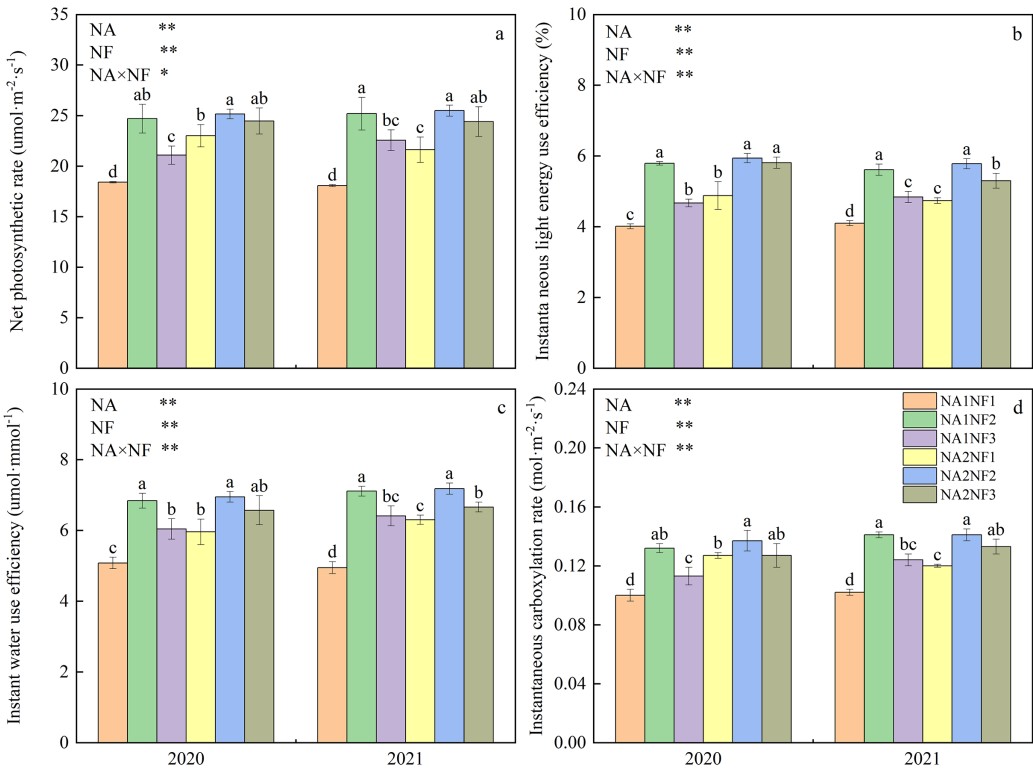

**Figure 2 Photosynthetic characteristics of maize at different fertilizer rates and nitrate-to-ammonium nitrogen (N) ratios.** Treatments: NA1NF1, N application rate of 210 kg·ha$^{-1}$ and nitrate-to-ammonium N ratio of 2:1; NA1NF2, N application rate of 210 kg·ha$^{-1}$ and nitrate-to-ammonium N ratio of 3:1; NA1NF3, N application rate of 210 kg·ha$^{-1}$ and nitrate-to-ammonium N ratio of 4:1; NA2NF1, N application rate of 300 kg·ha$^{-1}$ and nitrate-to-ammonium N ratio of 2:1; NA2NF2, N application rate of 300 kg·ha$^{-1}$ and nitrate-to-ammonium N ratio of 3:1; NA2NF3, N application rate of 300 kg·ha$^{-1}$ and nitrate-to-ammonium N ratio of 4:1. Bars and error bars indicate the mean + SE. Different lowercase letters above bars indicate a significant difference among treatments ($P = 0.05$, Tukey's HSD test). The single asterisk (*) in the table at the bottom left of the figure indicates a significant effect of the factor on the results ($P = 0.05$), and the double asterisks (**) indicate an extremely significant effect of the factor on the results ($P = 0.01$).     

activities were higher in NF2 than in NF1 by 23.07%, 20.22%, 14.43%, and 12.38%, respectively, in NA1 and by 19.23%, 20.65%, 10.58%, and 7.55%, respectively, in NA2. For NA1, the C metabolism indicators on DAA40 and DAA60 were significantly higher in NF2 than in NF3 in both years. At NA2, except for ME, all C metabolism indicators on DAA60 were significantly higher in NF2 than in NF3, and in 2021, post-anthesis indicators did not differ significantly between NF1 and NF3. For a given nitrate-to-ammonium N ratio, post-anthesis PEPC activity in NF1 was significantly higher in NA2 than in NA1 in both years (by an average of 7.08%). In 2021, all other indicators on DAA60 were significantly higher in NA2 than in NA1. No significant difference between NA1 and NA2 under the NF2 treatment was observed. On DAA60, the activities of ME, MDH, and PEPC in NF3 in NA1 did not differ significantly from those in NA2 in either year.

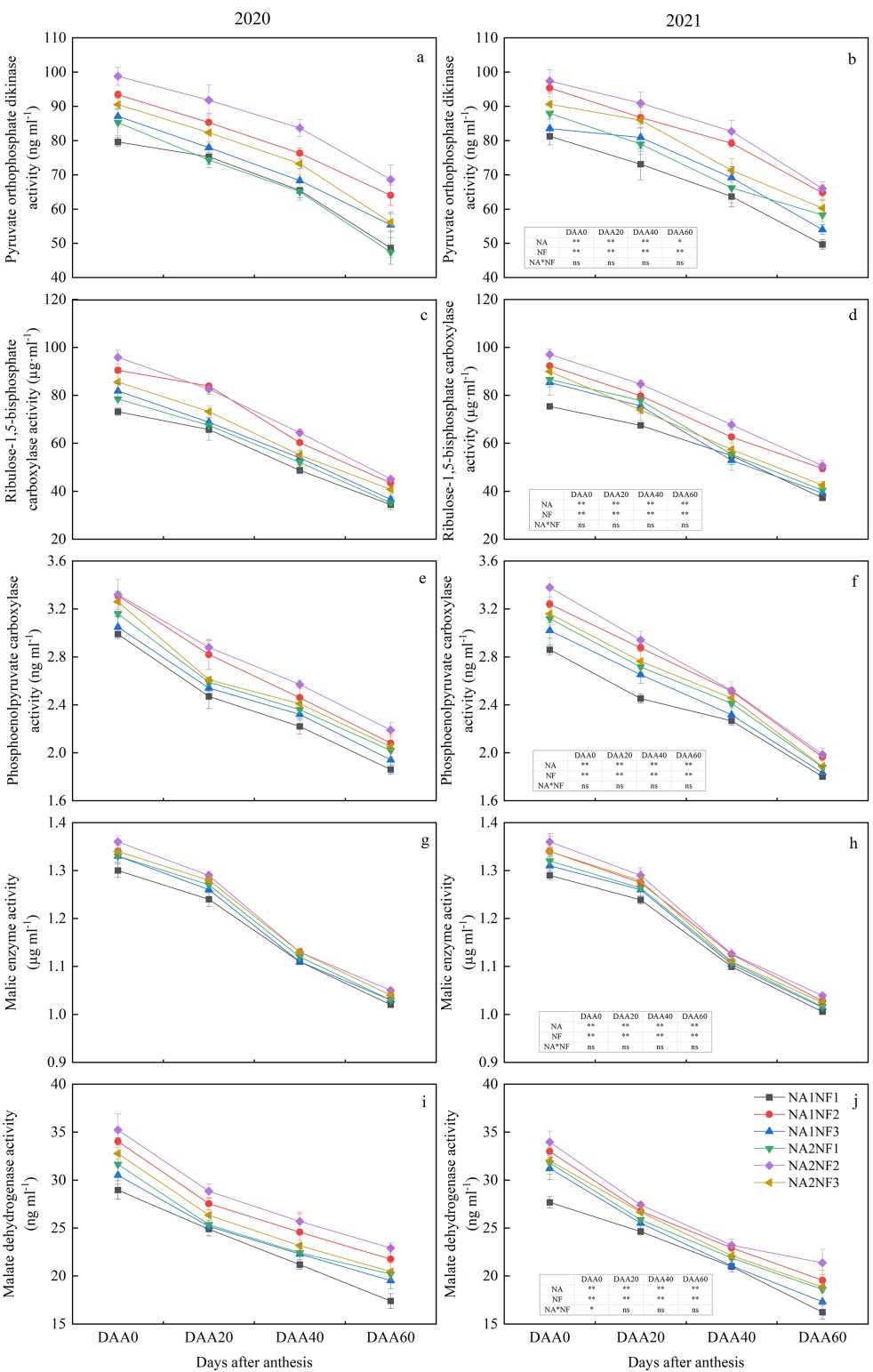

**Figure 3 Post-anthesis carbon metabolism enzyme activities in maize at different fertilizer rates and nitrate-to-ammonium nitrogen (N) ratios.** Treatments: NA1NF1, N application rate of 210 kg·ha$^{-1}$ and nitrate-to-ammonium N ratio of 2:1; NA1NF2, N application rate of 210 kg·ha$^{-1}$ and nitrate-to-ammonium N ratio of 3:1; NA1NF3, N application rate of 210 kg·ha$^{-1}$ and nitrate-to-ammonium N ratio of 4:1; NA2NF1, N application rate of 300 kg·ha$^{-1}$ and nitrate-to-ammonium N ratio of 2:1; NA2NF2,

**Figure 3** (continued)
N application rate of 300 kg·ha$^{-1}$ and nitrate-to-ammonium N ratio of 3:1; NA2NF3, N application rate of 300 kg·ha$^{-1}$ and nitrate-to-ammonium N ratio of 4:1. DAA0, 20, 40, and 60 represent the flowering period (anthesis), 20 days after anthesis, 40 days after anthesis, and the ripening period (60 days after anthesis). Error bars indicate the mean + SE ($P = 0.05$, Tukey's HSD test). The single asterisk (*) in the table at the bottom left of the figure indicates a significant effect of the factor on the results of a 2-year average ($P = 0.05$), and the double asterisks (**) indicate an extremely significant effect of the factor on the results ($P = 0.01$). The ns indicates the difference is not significant.

## Effects of nitrate-to-ammonium nitrogen ratio on maize post-anthesis nitrogen metabolism enzyme activities

Activities of enzymes involved in post-anthesis ear-leaf photosynthetic N metabolism decreased during growth and development in all treatments (Fig. 4). For a given N application rate, the post-anthesis N metabolism indicators were significantly higher in NF2 than in NF1 in both years. For NA1, average post-anthesis NR, GDH, GS, and GOGAT activities were higher in NF2 than in NF1 by 14.75%, 6.19%, 5.67%, and 8.29%, respectively. For NA2, average post-anthesis GOGAT and NR activities were higher in NF2 than in NF1 by 5.09% and 8.19%, respectively. On DAA40 and DAA60, all indicators were significantly higher in NF2 compared with those in NF3. The NR activity was higher in NF2 than in NF3 by 6.83% for NA1 and 6.79% for NA2. For NA2, the indicators in NF1 did not differ significantly from those in NF3. With the nitrate-to-ammonium N ratio of NF1, the flowering-stage indicators for NA2 were significantly higher than those for NA1 in both years. Activities of GDH and GOGAT for NA1 and NA2 did not differ significantly on DAA20 in 2020 and on DAA60 in 2021. Except for NR activity on DAA20 in 2020, the indicators in NF2 for NA1 and NA2 did not differ significantly. Activities of GDH and NR on DAA60 in NF3 for NA1 and NA2 did not differ significantly in either year.

## Effects of nitrate-to-ammonium nitrogen ratio on post-anthesis content of nonstructural carbohydrates in maize

Nitrogen application and nitrate-to-ammonium N ratio had extremely significant effects on the content of nonstructural carbohydrates in maize silking and maturity stages, and the interaction between the two factors was significant in maturity. The two factors also had extremely significant effects on transport, and the interaction was significant in sucrose and soluble sugar.

Ear-leaf-accumulated sucrose, soluble sugar, and starch in the silking stage were higher compared with those in the maturity stage (Table 2). For a given N application rate, except for sucrose and soluble sugar at NA2 in 2020, sucrose, soluble sugar, and starch accumulation in the silking stage were significantly higher in NF2 than in NF1 and NF3 in both years, and except for soluble sugar in 2020, the carbohydrate contents were significantly higher in NF3 than in NF1. At maturity stage, accumulated sucrose, soluble sugar, and starch were significantly higher in NF2 than in NF1 and NF3 for NA1 and were significantly higher than in NF1 for NA2 in both years. In 2021, sucrose, soluble sugar, and

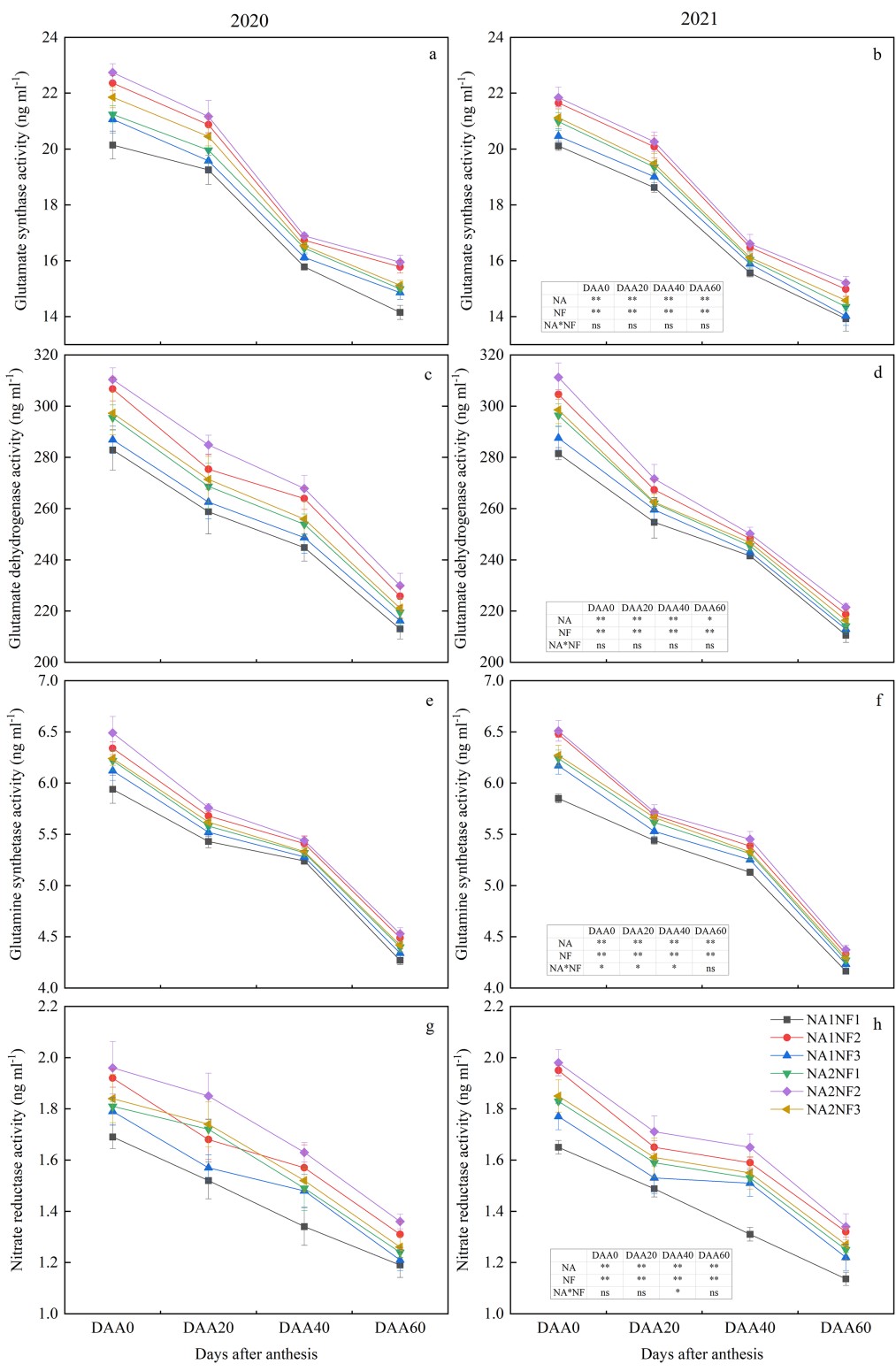

**Figure 4 Post-anthesis nitrogen (N) metabolism enzyme activities in maize at different fertilizer rates and nitrate-to-ammonium nitrogen (N) ratios.** Treatments: NA1NF1, N application rate of 210 kg·ha$^{-1}$ and nitrate-to-ammonium N ratio of 2:1; NA1NF2, N application rate of 210 kg·ha$^{-1}$ and nitrate-to-ammonium N ratio of 3:1; NA1NF3, N application rate of 210 kg·ha$^{-1}$ and nitrate-to-ammonium N ratio of 4:1; NA2NF1, N application rate of 300 kg·ha$^{-1}$ and nitrate-to-ammonium N ratio
**Figure 4** (continued)
of 2:1; NA2NF2, N application rate of 300 kg·ha$^{-1}$ and nitrate-to-ammonium N ratio of 3:1; NA2NF3, N application rate of 300 kg·ha$^{-1}$ and nitrate-to-ammonium N ratio of 4:1. DAA0, 20, 40, and 60 represent the flowering period (anthesis), 20 days after anthesis, 40 days after anthesis, and the ripening period (60 days after anthesis). Error bars indicate the mean + SE ($P = 0.05$, Tukey's HSD test). The single asterisk (*) in the table at the bottom left of the figure indicates a significant effect of the factor on the results of a 2-year average ($P = 0.05$), and the double asterisks (**) indicate an extremely significant effect of the factor on the results ($P = 0.01$). The ns indicates the difference is not significant.

**Table 2 Accumulated and transported nonstructural carbohydrates in maize at different fertilizer rates and nitrate-to-ammonium nitrogen (N) ratios.**

| Year | Treatment | Accumulation | | | | | | Transport | | |
|---|---|---|---|---|---|---|---|---|---|---|
| | | Silking stage | | | Maturity stage | | | | | |
| | | Sucrose | Soluble sugar | Starch | Sucrose | Soluble sugar | Starch | Sucrose | Soluble sugar | Starch |
| | | kg·ha$^{-1}$ | | | | | | | | |
| 2020 | NA1NF1 | 0.22c | 0.30d | 2.47d | 0.12d | 0.19d | 0.98e | 0.10d | 0.11d | 1.49c |
| | NA1NF2 | 0.35a | 0.44ab | 3.56ab | 0.19ab | 0.25ab | 1.57a | 0.16a | 0.19a | 1.99ab |
| | NA1NF3 | 0.27b | 0.32cd | 2.62c | 0.16c | 0.21c | 1.03d | 0.11c | 0.11d | 1.59bc |
| | NA2NF1 | 0.28b | 0.35bc | 2.73c | 0.17bc | 0.22bc | 1.01d | 0.11c | 0.13c | 1.72b |
| | NA2NF2 | 0.36a | 0.46a | 3.75a | 0.20a | 0.26a | 1.36ab | 0.16a | 0.20a | 2.39a |
| | NA2NF3 | 0.33a | 0.42ab | 3.54b | 0.19ab | 0.25ab | 1.28bc | 0.14b | 0.17b | 2.26a |
| 2021 | NA1NF1 | 0.21d | 0.26d | 2.20d | 0.11d | 0.14d | 0.82e | 0.10e | 0.12d | 1.38e |
| | NA1NF2 | 0.33ab | 0.41ab | 3.4ab | 0.19ab | 0.22ab | 1.50a | 0.19a | 0.17a | 1.9ab |
| | NA1NF3 | 0.24c | 0.29c | 2.29c | 0.13c | 0.15c | 0.86d | 0.11d | 0.14c | 1.43d |
| | NA2NF1 | 0.26c | 0.31c | 2.53c | 0.14c | 0.17c | 0.95c | 0.12c | 0.14c | 1.58c |
| | NA2NF2 | 0.35a | 0.45a | 3.58a | 0.20a | 0.28a | 1.47a | 0.15ab | 0.17a | 2.11a |
| | NA2NF3 | 0.31b | 0.39b | 3.25b | 0.17b | 0.23b | 1.23b | 0.14b | 0.16b | 2.02a |
| NA | | ** | ** | ** | ** | ** | ** | ** | ** | ** |
| NF | | ** | ** | ** | ** | ** | ** | ** | ** | ** |
| NA×NF | | ns | ns | ** | ** | ** | ** | ** | ** | ns |

**Note:**
Treatments: NA1NF1, N application rate of 210 kg·ha$^{-1}$ and nitrate-to-ammonium N ratio of 2:1; NA1NF2, N application rate of 210 kg·ha$^{-1}$ and nitrate-to-ammonium N ratio of 3:1; NA1NF3, N application rate of 210 kg·ha$^{-1}$ and nitrate-to-ammonium N ratio of 4:1; NA2NF1, N application rate of 300 kg·ha$^{-1}$ and nitrate-to-ammonium N ratio of 2:1; NA2NF2, N application rate of 300 kg·ha$^{-1}$ and nitrate-to-ammonium N ratio of 3:1; NA2NF3, N application rate of 300 kg·ha$^{-1}$ and nitrate-to-ammonium N ratio of 4:1. Different lowercase letters within a column indicate a significant difference between treatments ($P < 0.05$, Tukey's HSD test). The variance source is analyzed based on the average of each indicator in 2020 and 2021. The single asterisk (*) indicates a significant effect of the factor on the results ($P = 0.05$), the double asterisks (**) indicate an extremely significant effect of the factor on the results ($P = 0.01$), and the ns indicates the difference is not significant.

starch accumulation at maturity stage were significantly higher in NF2 than in NF3. Except for sucrose and soluble sugar with NA2 in 2020, sucrose, soluble sugar, and starch accumulation were significantly higher in NF2 than in NF1. Transported sucrose, soluble sugar, and starch were significantly higher in NF2 than in NF1. A total of 2-year average transported sucrose, soluble sugar, and starch were higher in NF2 than in NF1 by 57.14%, 56.52%, and 31.42%, respectively, for NA1, and by 34.78%, 37.04%, and 36.36%,

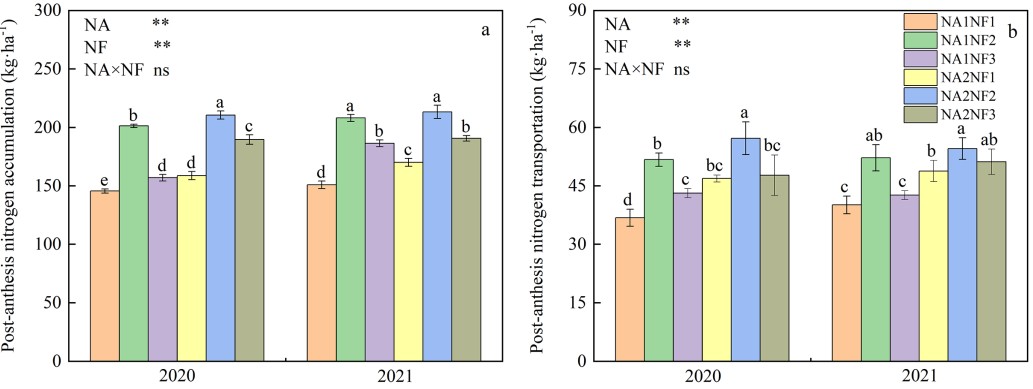

**Figure 5 Post-anthesis nitrogen (N) accumulation and transportation in maize at different fertilizer rates and nitrate-to-ammonium N ratios.** Treatments: NA1NF1, N application rate of 210 kg·ha⁻¹ and nitrate-to-ammonium N ratio of 2:1; NA1NF2, N application rate of 210 kg·ha⁻¹ and nitrate-to-ammonium N ratio of 3:1; NA1NF3, N application rate of 210 kg·ha⁻¹ and nitrate-to-ammonium N ratio of 4:1; NA2NF1, N application rate of 300 kg·ha⁻¹ and nitrate-to-ammonium N ratio of 2:1; NA2NF2, N application rate of 300 kg·ha⁻¹ and nitrate-to-ammonium N ratio of 3:1; NA2NF3, N application rate of 300 kg·ha⁻¹ and nitrate-to-ammonium N ratio of 4:1. Bars and error bars indicate the mean + SE. Different lowercase letters above bars indicate a significant difference among treatments ($P = 0.05$, Tukey's HSD test). The single asterisk (*) in the table at the bottom left of the figure indicates a significant effect of the factor on the results ($P = 0.05$), and the double asterisks (**) indicate an extremely significant effect of the factor on the results ($P = 0.01$). The ns indicates the difference is not significant.

respectively, for NA2. In 2021, transported soluble sugar and starch were significantly higher in NF3 than in NF1.

## Effects of nitrate-to-ammonium nitrogen ratio on post-anthesis nitrogen accumulation and transportation in maize

Nitrogen application and nitrate-to-ammonium N ratio had extremely significant effects on post-anthesis N accumulation and transportation in maize, but there was no interaction between the two factors. As shown in Fig. 5, under the same rate of N application, the accumulation of N at anthesis stage and N transfer after anthesis of maize were in the order of NF2 > NF3 > NF1, with significant differences among treatments; N accumulation at NF was 11.0–38.0% higher compared with that at NF1 and NF3. Under the NF2 N form ratio, N accumulation and transportation for NA2 treatment were significantly higher than those of NA1 in 2020, but the differences in these measures between N application rates in 2021 were not significant. Under the NF1 ratio, N accumulation and transportation were significantly higher for NA2 treatment compared with those of NA1 treatment in both years. Meanwhile, under the NF3 ratio, N accumulation was significantly higher for NA2 treatment than for NA1 treatment in 2020, and N transportation in 2021 was significantly higher for NA2 treatment compared with that of NA1 treatment.

## Effects of nitrate-to-ammonium nitrogen ratio on maize yield and nitrogen-use efficiency

Nitrogen application and nitrate-to-ammonium N ratio had extremely significant effects on maize yield, number of grains per ear, and weight per 1,000 grains. For both N

**Table 3 Maize yield and its components at different fertilizer rates and nitrate-to-ammonium nitrogen (N) ratios.**

| Year | Treatment | Effective number of ears (×10$^4$ ears ha$^{-1}$) | Number of grains per ear | Weight per 1,000 grains (g) | Yield (t ha$^{-1}$) |
|---|---|---|---|---|---|
| 2020 | NA1NF1 | 6.99a | 425.12c | 405.22c | 12.04e |
| | NA1NF2 | 7.10a | 434.92ab | 428ab | 13.22ab |
| | NA1NF3 | 6.97a | 428.90bc | 421.85b | 12.61cd |
| | NA2NF1 | 7.03a | 432.54bc | 408.74c | 12.42de |
| | NA2NF2 | 7.10a | 441.10a | 435.16a | 13.62a |
| | NA2NF3 | 7.08a | 434.72ab | 424.94ab | 13.08abc |
| 2021 | NA1NF1 | 7.03a | 418.03e | 402.31d | 11.82d |
| | NA1NF2 | 7.09a | 440.75ab | 428.35ab | 13.38ab |
| | NA1NF3 | 7.02a | 427.42d | 415.19c | 12.46c |
| | NA2NF1 | 6.98a | 432.60cd | 423.48bc | 12.79c |
| | NA2NF2 | 7.07a | 442.20a | 439.64a | 13.74a |
| | NA2NF3 | 7.06a | 435.94bc | 428.39ab | 13.18b |
| NA | | ** | ns | ** | ** |
| NF | | ** | ns | ** | ** |
| NA×NF | | ns | ns | ns | ns |

**Note:**
Treatments: NA1NF1, N application rate of 210 kg·ha$^{-1}$ and nitrate-to-ammonium N ratio of 2:1; NA1NF2, N application rate of 210 kg·ha$^{-1}$ and nitrate-to-ammonium N ratio of 3:1; NA1NF3, N application rate of 210 kg·ha$^{-1}$ and nitrate-to-ammonium N ratio of 4:1; NA2NF1, N application rate of 300 kg·ha$^{-1}$ and nitrate-to-ammonium N ratio of 2:1; NA2NF2, N application rate of 300 kg·ha$^{-1}$ and nitrate-to-ammonium N ratio of 3:1; NA2NF3, N application rate of 300 kg·ha$^{-1}$ and nitrate-to-ammonium N ratio of 4:1. Different lowercase letters within a column indicate a significant difference between treatments ($P < 0.05$, Tukey's HSD test). The variance source is analyzed based on the average of each indicator in 2020 and 2021. The single asterisk (*) indicates a significant effect of the factor on the results ($P = 0.05$), the double asterisks (**) indicate an extremely significant effect of the factor on the results ($P = 0.01$), and the ns indicates the difference is not significant.

application rates in 2020 and 2021, yields were significantly higher in NF2 and NF3 than in NF1 by 10.99% and 4.89%, respectively, in NA1 and by 7.22% and 4.05%, respectively, in NA2 (Table 3). Except for yield for NA2 in 2020, yield was significantly higher in NF2 compared with that in NF3. In 2021, the 1,000-grain weight was significantly higher in NF2 than in NF3 for NA1, and the number of grains per ear was significantly higher in NF2 than in NF3 for both NA1 and NA2. For any given nitrate-to-ammonium N ratio, the yield, the 1,000-grain weight, and the number of grains per ear in NA1 did not differ significantly from those in NA2 in 2020. In contrast, in 2021, only NF2 did not show a significant difference between NA1 and NA2 for these three measurements (yield, 1,000-grain weight, and number of grains per ear). Treatments did not differ significantly in the effective number of ears in either year.

Nitrogen application and nitrate-to-ammonium N ratio had extremely significant effects on N-use efficiency; however, the two factors only had significant influences on NAE. Table 4 shows that for a given nitrogen application rate, the differences between PFPN and nitrogen forms at NA2 were not significant in either year, and that the PFPN values for NF2 and NF3 at the NA1 rate in 2021 were not significantly different but were significantly higher compared with that of NF1 treatment. At the NA1 application rate, the differences in NAE for all N form ratios reached a significant level (NF2 > NF3 > NF1) in both years, while at the NA2 application rate, there was no significant difference in NAE between NF2 and NF3, but it was significantly higher compared with that in NF1

**Table 4 Nitrogen-use efficiency of maize at different fertilizer rates and nitrate-to-ammonium nitrogen (N) ratios.**

| Year | Treatment | PFPN (kg kg$^{-1}$) | NAE (kg kg$^{-1}$) |
|------|-----------|---------------------|--------------------|
| 2020 | NA1NF1 | 57.90a | 12.77c |
| | NA1NF2 | 63.10a | 17.96a |
| | NA1NF3 | 60.43a | 15.30b |
| | NA2NF1 | 41.60b | 10.01d |
| | NA2NF2 | 44.70b | 13.11bc |
| | NA2NF3 | 43.57b | 11.97cd |
| 2021 | NA1NF1 | 56.05b | 18.76c |
| | NA1NF2 | 63.38a | 26.10a |
| | NA1NF3 | 59.10ab | 21.81b |
| | NA2NF1 | 42.40c | 16.30d |
| | NA2NF2 | 45.37c | 19.27c |
| | NA2NF3 | 43.57c | 17.47cd |
| NA | | ** | ** |
| NF | | ** | ** |
| NA×NF | | ns | ns |

**Note:**

Treatments: NA1NF1, N application rate of 210 kg·ha$^{-1}$ and nitrate-to-ammonium N ratio of 2:1; NA1NF2, N application rate of 210 kg·ha$^{-1}$ and nitrate-to-ammonium N ratio of 3:1; NA1NF3, N application rate of 210 kg·ha$^{-1}$ and nitrate-to-ammonium N ratio of 4:1; NA2NF1, N application rate of 300 kg·ha$^{-1}$ and nitrate-to-ammonium N ratio of 2:1; NA2NF2, N application rate of 300 kg·ha$^{-1}$ and nitrate-to-ammonium N ratio of 3:1; NA2NF3, N application rate of 300 kg·ha$^{-1}$ and nitrate-to-ammonium N ratio of 4:1. Different lowercase letters within a column indicate a significant difference between treatments ($P < 0.05$, Tukey's HSD test). The variance source is analyzed based on the average of each indicator in 2020 and 2021. The single asterisk (*) indicates a significant effect of the factor on the results ($P = 0.05$), the double asterisks (**) indicate an extremely significant effect of the factor on the results ($P = 0.01$), and the ns indicates the difference is not significant.

treatment. PFPN and NAE were significantly higher in NA1 treatment compared with those in NA2 treatment.

## Correlation analysis of photosynthetic characteristics, post-anthesis carbon and nitrogen metabolic enzyme activity, non-structural carbohydrate and yield

As can be seen from Fig. 6, photosynthetic characteristics were significantly positively correlated with post-anthesis carbon and nitrogen metabolic enzyme activities and non-institutional carbohydrates, while post-anthesis carbon and nitrogen metabolic enzyme activities were significantly positively correlated with non-institutional carbohydrates. The yield was positively correlated with photosynthetic characteristics, metabolic enzyme activity in post-anthesis carbon belt and non-structural carbohydrate.

## DISCUSSION

Photosynthesis is the physiological and material basis of crop yield, and photosynthetic performance directly affects grain yield (*Shang et al., 2022*). Nitrate and ammonium N are forms of inorganic N that can be directly absorbed and used by crop root systems, and application rates and mixture ratios have major effects on photosynthetic performance. Nitrate and ammonium N uptake in maize depends on the stage of growth and

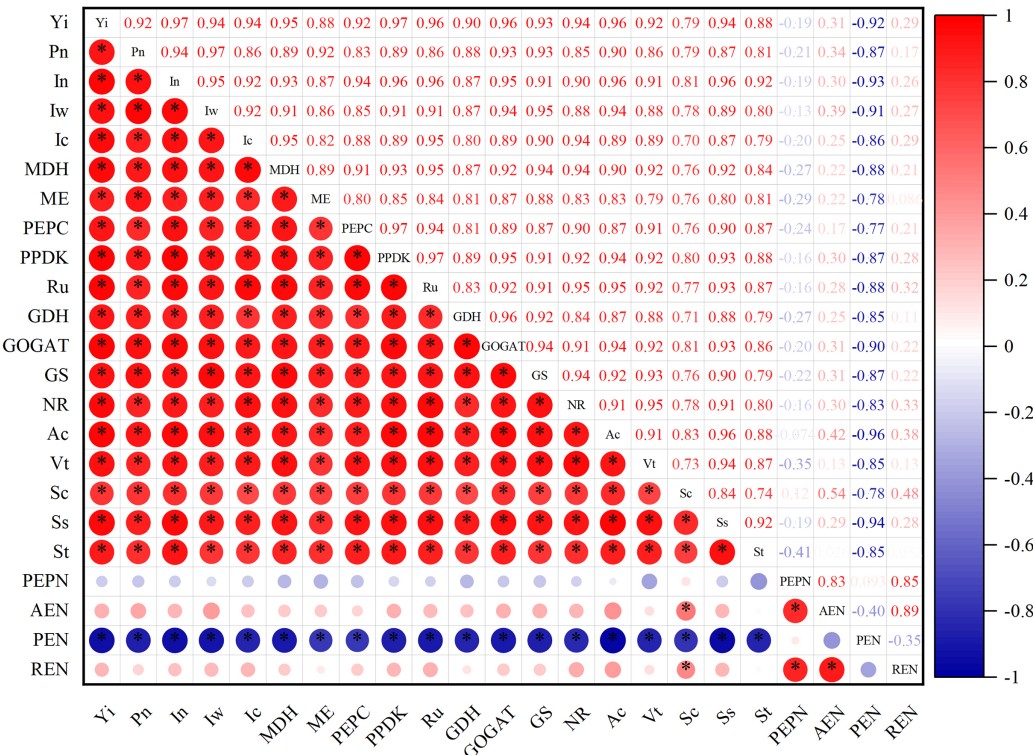

**Figure 6 Correlation analysis of photosynthetic characteristics, post-anthesis carbon and nitrogen metabolic enzyme activity, non-structural carbohydrate and yield.** The single asterisk (*) indicates a significant effect on the results of a 2-year average ($P = 0.05$), and the double asterisks (**) indicate an extremely significant effect on the results ($P = 0.01$).

development. In early stages of growth and development, N is predominantly absorbed in the form of ammonium, whereas in later stages, nitrate is the major form of absorbed N (*He, Li & Li, 1999*), suggesting that nitrate and ammonium N should be supplied at appropriate application rates at appropriate times. *Li et al. (2017)* reported that a nitrate-to-ammonium N ratio of 3:1 significantly increased leaf area and photosynthetic potential throughout the growth and development of maize, which resulted in a significantly improved photosynthetic performance. Our study produced similar conclusions. A nitrate-to-ammonium N ratio of 3:1 increased silking-stage photosynthetic performance, ear-leaf photosynthetic rate, instantaneous LEUE, instantaneous WUE, and instantaneous carboxylation rate compared with those at ratios of 2:1 and 4:1. An excessively high nitrate-to-ammonium N ratio in the silking stage leads to excess nitrate N and insufficient ammonium N in maize, thereby suppressing growth. In contrast, an excessively low nitrate-to-ammonium N ratio leads to excess ammonium N that damages biomembranes, reduces oxidative phosphorylation, uncouples photosynthetic and nonphotosynthetic phosphorylation, and reduces ATP generation, thereby reducing C fixation from $CO_2$ and the photosynthetic rate in maize (*Li et al., 2018*). Thus, appropriate nitrate and ammonium N application rates help to maintain substantial photosynthetic performance in maize.

Carbon and N metabolism in plants is important for the fixation of light energy, the synthesis of energy substances, the conversion and accumulation of carbohydrates, and the

conversion of inorganic N (*Hu et al., 2021*). The activity of key enzymes involved in C and N metabolic pathways plays a key role in the process of C and N metabolism in plants. The assimilation of C and N in plants is more than that in leaves through photosynthesis. The activities of enzymes related to carbon metabolism, such as 1, 5-diphosphate ribulose carboxylase, pyruvate phosphate dikinase and malate dehydrogenase, and the activities of enzymes related to nitrogen metabolism, such as nitric reductase, glutamate dehydrogenase and glutamine synthetase are enhanced, which is conducive to better photosynthesis in leaves. Increase the photosynthetic rate and promote the assimilation of C and N in plants. And, these enzymes respond differently to different application rates and forms of N (*Ben Mrid et al., 2018*; *Wu et al., 2019*). The activities of NR and GS enzymes in maize leaves could be improved by increasing nitrogen application rate. But, *Wang et al. (2009)* observed that the activities of NR, GS, and sucrose phosphate synthetase in maize leaves were markedly higher at an N application rate of 300 kg·ha$^{-1}$ compared with those at an N application rate of 450 kg·ha$^{-1}$, indicating that an excessively high N application rate reduces the activities of enzymes involved in C and N metabolism. *Li et al. (2021)* observed that, compared with other ratios, a nitrate-to-ammonium N ratio of 2:1 increased the activities of NR, GS, and GDH in the leaves of sweet potato (*Ipomoea batatas*). *Wang et al. (2014)* reported that, for a given N application rate, a nitrate-to-ammonium N ratio of 3:1 increased C and N metabolism in *Ginkgo biloba*, compared with that under other ratios. In this study, at a nitrate-to-ammonium N ratio of 3:1, the N application rates of 210 and 300 kg·ha$^{-1}$ had no significant effects on C and N metabolism and the key enzyme activities of non-structural carbohydrate accumulation and transport in maize. In this study, at a nitrate-to-ammonium N ratio of 3:1, the activities of key enzymes involved in C and N metabolism and nonstructural carbohydrate accumulation and transport in maize a did not differ significantly between N application rates of 210 and 300 kg·ha$^{-1}$. The similarity in responses at different N application rates might be because the technique of shallow-buried drip irrigation with water–fertilizer integration promoted the uptake and utilization of N by maize roots. In addition, the supplementary nitrogen fertilizer in the study was in the form of ammonium nitrogen and nitrate nitrogen, which could be directly and quickly absorbed and utilized, and thus had a higher nitrogen use efficiency (*Chen et al., 2021*; *Xue, Yang & Zhang, 2022*). The N application rate of 210 kg·ha$^{-1}$ likely satisfied the N requirement for normal C and N metabolism, whereas the N application rate of 300 kg·ha$^{-1}$ might have led to excess N and stressed the maize root system. An excess supply of nitrate and ammonium N caused physiological stress in the maize leaves or that a high concentration of $NH_4^+$-N or $NO_3^-$-N reduced the transport of $NH_4^+$ and $NO_3^-$ from the roots to the leaves (*Seck-Mbengue et al., 2017*), and reduced the activities of enzymes related to carbon and nitrogen metabolism. Consequently, the intensity of C and N metabolism and grain yield did not increase as the N application rate increased. On the contrary, the lower nitrogen application rate of 210 kg·ha$^{-1}$ did not decrease, and the higher nitrogen use efficiency was obtained.

The supply of C and N to maize seeds can be satisfied by C assimilation in leaves, N uptake by the root system, and transport of C and N from vegetative parts to seeds (*Miao et al., 2010*). Thus, improving the intensity and coordination of C and N metabolism in

maize is particularly important for improving yield (*Chang et al., 2017*). Nitrogen can promote the transport of C to grain (*Vijayalakshmi et al., 2013*; *Ma et al., 2023*). An insufficient N supply decreases the 1,000-grain weight, whereas an excessive N supply leads to a decrease in C-to-N ratio and excessive metabolism in leaves, thereby reducing the export of photosynthetic products and adversely affecting seed formation. In this study, at a nitrate-to-ammonium N ratio of 3:1, maize yields for N application rates of 210 and 300 kg·ha$^{-1}$ did not differ significantly over 2 years, which was consistent with the variation in photosynthetic characteristics, post-anthesis enzyme activities of C and N metabolism. Through correlation analysis, it was further demonstrated that the yield changes were closely related to photosynthetic characteristics, post-anthesis C and N metabolic enzymes, and non-structural carbohydrate accumulation and transport. In addition, compared with the conventional nitrogen (urea) application, the yield and nitrogen use efficiency increased by 6.80% and 6.79% under the same water and nitrogen dosage. The reason for the high yield and nitrogen use efficiency is that nitrogen fertilizer supplementation with appropriate amount and appropriate ratio of nitrate nitrogen and ammonium nitrogen can be directly absorbed and utilized by maize, which can promote protein production, accelerate the conversion and utilization of sucrose, starch and soluble sugar, and increase the metabolism of C and the synthesis of corresponding products (*Wang et al., 2019b*) Carbon metabolism provides the necessary energy and C skeletons for reduction of $NO_3^-$ and synthesis of amino acids and, therefore, improving C metabolism increases N metabolism enzyme activities, assimilation of $NH_4^+$, and efficiency of N metabolism in crops (*Zhang et al., 2021*). Therefore, appropriate nitrate, ammonium and nitrogen ratio and nitrogen application amount can facilitate rapid absorption with maize roots, promote photosynthesis, improve maize C and N metabolism, and enhance the transport of non-structural carbohydrates to grains, which is the major reason for the increases in yield and N fertilizer efficiency of maize in this study.

## CONCLUSIONS

In this study, at relatively low nitrogen application rate, we discussed that application of suitable combination of nitrate and ammonium N would change C and N metabolism in maize, thereby improving maize photosynthesis and ultimately achieving synergistic improvement of maize N content, yield, and N efficiency. Among the tested treatments, a N application rate of 210 kg·ha$^{-1}$ and a nitrate-to-ammonium N ratio of 3:1 enhanced the silking-stage photosynthetic rate, instantaneous LEUE, instantaneous WUE, and C and N metabolism enzyme activities, promoted the accumulation and transport of nonstructural carbohydrates and post-anthesis N, and synergistically increased maize yield and N-use efficiency. These findings provide theoretical guidance on appropriate N applications for high-yield maize cultivation.

## ACKNOWLEDGEMENTS

We thank the reviewers for their careful, constructive, and insightful comments on the manuscript. In addition, we thank Charlesworth Author Services for editing the English language of a draft of this article.

### Funding

This work was supported by the National Natural Science Foundation of China (No. 32160509) and the Natural Science Foundation of Inner Mongolia (No. 2021BS03005). The funders had no role in study design, data collection and analysis, decision to publish, or preparation of the manuscript.

### Grant Disclosures

The following grant information was disclosed by the authors:
National Natural Science Foundation of China: 32160509.
Natural Science Foundation of Inner Mongolia: 2021BS03005.

### Competing Interests

The authors declare that they have no competing interests.

### Author Contributions

- Fanhao Meng conceived and designed the experiments, performed the experiments, analyzed the data, prepared figures and/or tables, authored or reviewed drafts of the article, and approved the final draft.
- Ruifu Zhang conceived and designed the experiments, authored or reviewed drafts of the article, and approved the final draft.
- Yuqin Zhang conceived and designed the experiments, authored or reviewed drafts of the article, and approved the final draft.
- Weimin Li performed the experiments, prepared figures and/or tables, and approved the final draft.
- Yushan Zhang performed the experiments, prepared figures and/or tables, and approved the final draft.
- Mingwei Zhang performed the experiments, prepared figures and/or tables, and approved the final draft.
- Xuezhen Yang performed the experiments, analyzed the data, prepared figures and/or tables, and approved the final draft.
- Hengshan Yang conceived and designed the experiments, analyzed the data, authored or reviewed drafts of the article, and approved the final draft.

### Data Availability

  The raw measurements are available in the Supplemental File.

### Supplemental Information

Supplemental information for this article can be found online at http://dx.doi.org/10.7717/peerj.16548#supplemental-information.

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
