# Peer review of "Improving maize carbon and nitrogen metabolic pathways and yield with nitrogen application rate and nitrogen forms"

_PeerJ, doi:10.7717/peerj.16548_

## Round 0.1 · original submission · Major Revisions

Dear Dr. Meng,
Your work has been assessed by two independent experts. They both agreed that the paper could be published in PeerJ, but it should be significantly revised beforehand. Please read the comments of the reviewers and respond to all of them.
With best regards,

**Language Note:** PeerJ staff have identified that the English language needs to be improved. When you prepare your next revision, please either (i) have a colleague who is proficient in English and familiar with the subject matter review your manuscript, or (ii) contact a professional editing service to review your manuscript. PeerJ can provide language editing services - you can contact us at [email protected] for pricing (be sure to provide your manuscript number and title). – PeerJ Staff

Reviewer 1 ·

Basic reporting

The manuscript entitled ”Improving maize carbon and nitrogen metabolic
pathways and yield with nitrogen application rate
and nitrate-to-ammonium nitrogen ratio” is well designed and analysis is mostly reasonable.
However, please revise the following points.

1) In the abstract line 27, Two rates of N application [210 kg·ha21 (N1)
28 and 300 kg·ha21 (N2)], but in the Experimental design on Line 95 the N application rates are two
95 levels: 200 kg·ha21 (N1) and 300 kg·ha21 (N2).

Experimental design

1) Line 108; the planting density of 750,000 plants ha-1, is equivalent with 75 plants m-2. This seems too high, and 7.5 plants m-2 may be appropriate from Table 3 the effective number of ear about 7.

2) In Figure 3, the chart axis label of the Pyruvate orthophosphate dikinase gene, the "gene" should be deleted.

3) In Figure 5, the chart axis label unit is "hm-2", although the other part " ha" is used. I think you can use ha in this part.

Validity of the findings

This article is valid for publication to meet the standard.

Please add the discussion part about the yield of conventional cultivation using urea in this area.
Because the authors did not compare conventional cultivation with urea.

Additional comments

No additional comments

Reviewer 2 ·

Basic reporting

See below

Experimental design

See below

Validity of the findings

See below

Additional comments

The research investigates the effect of mixed N supply on maize productivity in the field and explain the underlying physiological mechanism. Although there are many research on this topic, few research are done in field conditions. These research show that mixed supply of nitrate and ammonium in the ratio of 3:1 can significantly increase maize productivity and NUE in the field and this effect is related to improved photosynthesis and efficient C and N utilization in the plant. The data is relatively complete. Yet, there are many problems. The paper needs to be greatly modified before it can be accepted for publication.


Major points:
Grain yield is not only determined by Pn but also by LAD. Ammonium can have significant effect on leaf growth. The higher Pn may be related to leaf senescence. Were LAI and leaf senescence determined?

Why is Pn determined? Does N forms specially affect Pn? The relative information should be introduced in Introduction.

Why were these enzymes determined? how do these enzymes contribute to Pn? Are these enzymes affected by N forms in the previous studies? The related information should be introduced in Intorduction.

The statistical method is not well explained. Which statistical program is used to analyze the effect of N amount, N form and year effect? Two way? Three-way?

In Results, some analysis may be conducted to analyze the relationship between the parameters, and the contribution of one parameter to the other parameter, and finally, grain yield.

The description of the results was not much logic. the author should clearly state the effect of nitrogen, and then nitrogen forms.

The logic of Discussion is not good. there is not a linkage between grain yield, yield component, dry matter production and re-translocation, N accumulation and remobilization, and the enzymes. Each paragraph is independent. But the paper did not finally answer the question "how did P2 treatment increase maize yield". That is, the relationship between maize yield and the other parameters is not established.


There are many papers relating to N form effect on maize growth and the underlying physiological mechanism. However, few papers in maize are cited.


Minor questions:
Title should be modified to stress the effect of N forms.
In Abstract, no N treatment is not mentioned. N amount of 210 is not consistent with 200 in M&M.
line 96. The abbreviation of "P" is not good, because it is normally used for "Phosphorus" in fertilizer experiment. "N amount", "N form" may be used for the two treatment.
Line 111. Why was Pn determined only at silking stage, while the enzyme activity was determined through the grain filling stage?At which time during the day was Pn determined? what were the light intensity, leaf temperature, CO2 conditions?
Line 122, the methods for enzyme determination should be introduced in brief.
Line 133, the methods for determination fo NSC should be introduced in brief.
line 139, transportation should be "remobilization"
line 147, N content should be N concentration.
line 149, "ripening stage" should be "maturity"
line 154. The calculation of "physiological N use efficiency" is not reasonable. it should be calculated as grain yield / total N accumulation. It is impossible to distinguish the role of N derived from the soil and fertilizer.
line 161, "number of grains per ear" should be calculated for grain yield, effective ear number, and grain weight. the counting result from ten sampling ears is only an estimation, and not reliable.
line 176. English writing need to be improved. here, the two years' results is not described properly.
Line 277, in Discussion, the physiological mechanism of the N form on maize grain yield under field level should be discussed in more detail.
line 299, Shallow-buried drip irrigation is only a way to apply N form treatments. It is not necessary to be discussed.
line 311, the exact relationship between each enzyme to Pn should be discussed in detail. their possible contribution should be stressed.
line 331, the effect of N amount on enzyme activity, Pn, grain yield should not be discussed independently. These effect should be discussed in the context of NUE. That is, Why did P2 treatment increased grain yield with a reduced N input.
Figures 2-5, the symbol of each treatment should be formatted in a way that one easily distinguishes the treatment of N amount and N forms. Each sub-figure should be named, such as 2a, 2b.
Fig 1 and Fig5, the significance of N , P and N x P is shown only for 2020. How about the year 2021.
Figure 2-4, significance test is lacking. Is there an interaction effect between N amount and N forms? if not, then the data between N amount should be averaged and just show the effect of N forms.
Figure 3 and 4. “activity” should be added in the description of Y-axis.
In all tables, variance source should be added to show the effect of single factor and the interaction.
Table 1. leaf age should be given to stead of “shooting stage, big-flare stage”.

---

## Round 0.2 · Minor Revisions

Dear Dr. Meng,

Two independent reviewers re-evaluated your work. They both agree that most of their comments were taken into account by the authors. However, they both still have some minor concerns. Please respond to them in the cover letter.

Please note that the second reviewer suggests that you cite two works. You can do this if you think it is justified. However, if you feel that these works should not be cited by you, this will not influence my decision to accept the work.
With best regards,

Reviewer 1 ·

Basic reporting

The authors revised the errors, and it is much improved.

Figure 1 needs revision.
Precipotation should be changed to Daily precipitation (mm/day).
Air temperature should be changed to Average air temperature.

The unit of Figure
is umol, but this is micro mol.

Experimental design

Experimental design is OK.

Validity of the findings

Impact of this paper is an average. But this experiment was done from a new approach.

Reviewer 2 ·

Basic reporting

The manuscript has greatly improved according to the suggestions. But there are still some problems:

Line 162-163: the formula is not accurate. I still suggest that the word “transportation” be replaced with “remobilization” throughout the manuscript, because it indicates the amount of N remobilized from the vegetative organs to the grains during post-silking stage. Here, it includes two processes, protein degradation and N translocation. The word “transportation” only indicates a simple transport of something, for example in the xylem, is not correct to indicate the meaning that the authors want to express. And the accurate formula should be “ N Remobilization = N accumulation in the stalk at maturity / N accumulation in the stalk at silking stage ”.

Line 164: “Nitrogen partial productivity” should be revised as “partial fertilizer productivity of N (PFPN)”

Line 167-line 169, Again, I strongly suggest to delete this parameter. The word “physiological nitrogen use efficiency” is normally calculated as “grain yield / total N accumulation”, indicating that grain production per unit of plant N accumulation. Although someone uses the same formula as the authors used to indicate so-called “physiological N use efficiency”, the calculating result is misleading, and cannot be explained in physiology.

Finally, the authors may like to add the following references to understand the physiological basis of N form effect on maize yield.

Wang P, et al. 2019. Increased biomass accumulation in maize grown in mixed nitrogen supply is mediated by auxin synthesis. Journal of Experimental Botany, 70: 1859–1873.
Wang P.et al. 2019. Interaction effect of nitrogen form and planting density on plant growth and nutrient uptake in maize seedlings. Journal of Integrative Agriculture 2019, 18(5): 1120–1129

Experimental design

OK

Validity of the findings

OK

Additional comments

NO

---

## Round 0.3 · accepted · Accept

Dear Dr. Meng,

Both reviewers agreed that the work could be accepted in its current version. My congratulations!

With best regards,

Reviewer 1 ·

Basic reporting

The manuscript has been well revised.

Experimental design

Experimental design is good.

Validity of the findings

The finding of valid.

Additional comments

The unit of the vertical axis in Figure 2 (a) and (c) is still not appropriate.
"umol" should be changed to "micro mol".

Reviewer 2 ·

Basic reporting

Sorry for a typing mistake in my suggestions in calculating the parameter “N remobilization”. The right formula should be: N Remobilization = N accumulation in the stalk at silking stage - N accumulation in the stalk at maturity. Please check the manuscript to see whether or not this mistake affected the data analysis.

I have no other suggestions.

Experimental design

no comment

Validity of the findings

no comment